# Primary Non-Hodgkin Uterine Lymphoma of the Cervix: A Literature Review

**DOI:** 10.3390/medicina58010106

**Published:** 2022-01-10

**Authors:** Cristina Capsa, Laura Aifer Calustian, Sabina Antonela Antoniu, Eugen Bratucu, Laurentiu Simion, Virgiliu-Mihail Prunoiu

**Affiliations:** 1Department of Radiology “Prof. Dr. Alexandru Trestioreanu”, 022328 Bucharest, Romania; cristinacapsa@yahoo.com; 2Medical Oncology II Oncological Institute “Prof. Dr. Alexandru Trestioreanu”, Carol Davila University of Medicine and Pharmacy, 022328 Bucharest, Romania; laura_calustian@yahoo.com; 3Department of Medicine II, University of Medicine and Pharmacy Grigore T Popa, 700115 Iasi, Romania; sabina.antoniu@outlook.com; 4Clinic I General and Oncological Surgery “Prof. Dr. Alexandru Trestioreanu”, Carol Davila University of Medicine and Pharmacy, 022328 Bucharest, Romania; bratucueugen@yahoo.com (E.B.); laurentiu.simion@umfcd.ro (L.S.)

**Keywords:** cervical tumour, primitive non-Hodgkin large B-cell, cervical lymphoma

## Abstract

*Introduction:* Non-Hodgkin lymphomas (NHL) comprise 85% of the total lymphomas diagnosed, with the histological type of diffuse large B-cell lymphomas (DLBCL) being the most prevalent in adults. In about 40% of cases, the location is extranodal. Uterine cervix lymphomas of this type are extremely rare (0.5–1.5%) and represent a diagnostic challenge. A case of DLBCL of the cervix is presented here along with a review of the literature. *Materials and methods:* A 75-year-old patient was referred with a bleeding vegetant tumour occupying her entire vagina. The histological and pathological investigations performed following the tumour biopsy indicated a malignant, diffuse, vaguely nodular lymphoid tumour proliferation. The immunohistochemistry results were in favour of a diffuse B-cell non-Hodgkin lymphoma (DLBCL). CHOP (Cyclophosphamide, Hydroxydaunorubicin (also called doxorubicin or adriamycin), Oncovin (vincristine), Prednisone or Prednisolone) polychemotherapy and radiotherapy were effective and resulted in tumour regression (from 3.4 cm to tumour disappearance, with the cervix returning to normal size). *Conclusions:* The uterine cervix lymphoma prognosis is more conservative than that for a nodal lymphoma, mainly due to a later diagnosis determined via immunohistochemistry. Chemotherapy is the main treatment.

## 1. Introduction

Diffuse large B-cell lymphomas (DLBCL) are the lymphoid neoplasms most frequently diagnosed in adults (about 40%) every year [1].

Approximately 40% of diffuse large B-cell lymphomas are extranodal; the gastrointestinal tract (15%), bone, and central nervous system are the most common sites involved [2,3]. Uterine cervix lymphoma, however, is very rare; for example, in 2016, 246 cases were reported in the literature, with about 75% being primary (strictly localised to the cervix) and the remaining cases being secondary [2,3].

The clinical symptoms of uterine cervix lymphoma are nonspecific to the histological type and may include vaginal bleeding, vaginal tumour, dyspareunia, and pelvic pain [2,3].

We report a case of primary cervix DLBCL in an elderly female, discuss the related diagnostic work up, and compare our results with those reported in the literature.

## 2. Methods and Results

A 75-year-old patient was referred to our clinic in 2018 with significant vaginal bleeding and a diagnosis of cervical tumour. The gynaecological exam described the presence of a bleeding vegetant tumour occupying the entire vagina up to the introitus; it was hard to delineate the uterus, which had infiltrated perimeters, with the left one next to the bone wall. Anaemia (haemoglobin level of 9.47 g/dL) and leucocytosis (10,600 cells/mmc) were also present. A cervical biopsy was performed, and diffuse malignant lymphoid tumour proliferation was detected: vaguely nodular, with large–medium polymorphous cells. An immunohistochemistry (IHC) study was afterwards performed, and a histological diagnosis of non-Hodgkin large B-cell malignant lymphoma (DLBCL) NOS (Not Otherwise Specified), non-GC (non-germinal center subgroup) (was established based on the following immune patterns: large B-cell tumour proliferation, CD20 and BCL2 positive, AE1-AE3 negative, p63 negative in atypical large cells (ACL), CD10 negative, BCL6 positive, Mum1 positive, p16 positive, CD5 negative, Ki67: 50% in atypical large cells. The patient was subsequently referred to the haematologist and a bone marrow biopsy was performed; no evidence of marrow involvement was found after IHC studies. All procedures performed were in accordance with the ethical standards of the 1964 Helsinki Declaration and its later amendments.

Whole-body imaging with CT (Computed Tomography) scan detected no secondary lesions in the solid organs or in the lymph nodes except for inguinal bilateral adenopathies less than 1 centimetre in diameter. The primary tumour was represented by an intensely iodophilic, dense vaginal mass of 3.4/2.9 cm with no changes of the uterus or the uterine adnexa. The MRI (Magnetic resonance imaging) scan confirmed the primary cervical tumour (Figure 1).

The diagnosis of stage IE DLBCL of the cervix was established and the patient was referred to the haematology department, where the decision was made to treat her with the standard dose of CHOP (cyclophosphamide, doxorubicin, vincristine, and prednisone) every 3 weeks for six cycles. CHOP chemotherapy treatment was decided via a multidisciplinary consultation: doxorubicin 50 mg/m^2^ at day 1, vincristine 2 mg/m^2^ at day 1, cyclophosphamide 600 mg/m^2^ at day 1, and dexamethasone 16 mg/m^2^ from day 1 to day 5. Treatment was given every 21 days [1] for six courses of treatment. Although standard treatment is with R-CHOP every 3 weeks for six cycles, rituximab (R) (monoclonal antibody, 375 mg/m^2^) [1] was not considered due to chronic hepatitis C (diagnosis HCV-RNA = 134,000 i.u./L). A follow-up was carried out in 2019 at the end of chemotherapy, 8 months after the initial diagnosis: the general health status of the patient was fine; anaemia was absent and the gynaecological examination showed a significant size reduction of the tumour and no bleeding, with the cervix being of normal size. A post-treatment PET-CT was performed and this detected a metabolically active lesion in the cervix consistent with lymphomatous involvement (score 4 scale of Deauville five-point scale) [1] (Figure 2). Multidisciplinary consultation resulted in the recommendation of consolidation radiotherapy of the cervix and pelvic region; 45 Gy was administered over 25 fractions/5 weeks.

Two contrast-enhanced chest/abdomen/pelvis CT scans were performed in 2020 (at 12 and 18 months after radiotherapy): the cervix showed an apparently homogenous structure with no regional adenopathies and again no secondary lesions were detected (Figure 3). The vaginal clinical exam of the pelvis indicated a normal aspect. A surveillance PET-CT scan which was performed at the end of 2020 confirmed the patient’s favourable course (score 2 of scale of Deauville five-point scale) [1]. At the time of writing, i.e., 29 months after her initial diagnosis, the patient is alive, with a good status of health and no signs of a local recurrence.

## 3. Discussions and Literature Review

Uterine cervix lymphomas account for 0.008% of malignant cervical tumours [3]. Cervical lymphoma management is not commonly described in the literature because such cases rarely occur. In terms of clinical presentation, that of our case was a common one, represented by the presence of vaginal tumour and vaginal bleeding. Interestingly, and probably more evocative of a primary type of lymphoma, was the absence of pelvic pain, which is commonly reported in the literature, usually in secondary cases [3,4,5,6]. In our case, it was immunohistochemistry which confirmed the final diagnosis. 

The case presented herein is an Ann Arbor stage IE [6,7]. Studies indicate that 5-year survival is better in the IE stage (89%) than in the IIE–IV stages (20%) [2].

A PUBMED literature search for papers published within the past 5 years (2016–2020) identified 14 more cases of primary uterine cervix lymphomas (Table 1). The patient age varied between 31 and 79 years old and the average age of the group was 49. All patients were treated with R-CHOP chemotherapy (rituximab–cyclophosphamide, doxorubicin, vincristine, and prednisone) as their main treatment, and the majority had a positive evolution, with clinically and PET-CT- or CT-MRI-confirmed tumour disappearance and a disease-free interval between 12 and 48 months. Two patients (18.2%) were subjected to full hysterectomy, and another two (18.2%) were treated with pelvic radiotherapy as an adjuvant to the chemotherapy or as a measure against a cervix-located recidivation. We must note the prevalence of the R-CHOP chemotherapy as a standard treatment, with rituximab being removed from the therapy under very specific conditions (e.g., chronic hepatitis, etc.), with radiotherapy as an adjuvant treatment and surgery (full hysterectomy) being used in a small number of pre- or post-chemotherapy cases [8,9,10]. According to the therapeutic guidelines (e.g., NCCN), antiviral therapy for HCV can be used before starting chemoimmunotherapy and can be integrated into the treatment plan [1]. Adjuvant radiotherapy was recently introduced as a therapeutic option (2009) with the R-CHOP treatment as the standard, thus obtaining long-term remission [8,10]. Radiotherapy (RT), and especially advanced radiotherapy (IMRT—Intensity Modulated Radiation Therapy, etc.), is recommended as an adjuvant to chemotherapy, and works by targeting the affected organ and the proximal adenopathy areas [1]. Therefore, it can be used for tumours over 7.5 cm in diameter, chemotherapy-resistant tumours, and tumours with partial response to chemotherapy, or for patients with complications due to chemotherapy [4].

The immunohistochemistry markers subclassified DLBCL into two subtypes: germinal centre cell lymphoma (GC-B), with CD5-; CD10+; BCL6+ or CD5-; CD10-; BCL6+, IRF4/MUM1-; and the non-GC-B subtype: CD5-; CD10-; BCL6+, IRF4/MUM1+ (1).

These studies also highlight the importance of immunohistochemistry (IHC) as the most important diagnostic tool in the diagnosis and differential diagnosis of primary malignant cervical lymphomas. Currently, there is no IHC marker standard for this diagnosis. LCA, vimentin, CD20, CD30, Bcl6, and Bcl2 can be used, as reported in [2,8,9]. The differential diagnosis is made against chronic cervicitis, small-cell carcinomas, adenocarcinomas, and cervix sarcomas. The immunophenotype (MUM-1+, Bcl-6+, CD10−) is compatible with DLBCL non-GC-type. Immunohistochemistry also has prognostic value, being included in the Hans algorithm (CD10, BCL-6, and MUM1). Algorithms developed by Hans et al. using immunohistochemistry were first widely accepted as a mechanism to divide DLBCL into germinal centre (GC) and non-GC subtypes. The algorithm is based on IHC expressions of CD10 (Cluster of Differentiation 10), BCL6 (B Cell Lymphoma 6), and MUM1 (Multiple Myeloma 1) proteins. The advantage of using the Hans IHC algorithm is that it uses only three easily assessable antibodies, which makes it widely acceptable compared to other algorithms that were developed later to subtype DLBCL according to the cell of origin [8,9,10]. The literature also indicates that the recommended treatment in 62% of the cases is CHOP, with or without rituximab [2,8]. Monitoring the disease over, on average, 38 months demonstrated that only 19% of the patients had a reoccurrence (stages II–IV) and that the 5-year survival rate was 80% in the incipient stage, as in our case. Apart from age, disease stage, and immunophenotype, the extent of nodal involvement and the performance status were also cited in the literature as prognostic factors [2]. In our case, only age seemed to be a negative prognostic factor, but its contribution seemed to be negligible, given the favourable course on a medium-term basis.

As this is a rare disease, there is no consensus regarding the best treatment. Lymphomas are considered chemotherapy- and radiotherapy-sensitive [19] and, therefore, surgical treatment is limited. Surgery is mostly used for uterine cervix biopsy and conisation, and hysterectomies are rare [2,8,9]. A recommended treatment might be six rounds of R-CHOP + RT (45Gy) [11,20]. The average tumour size is 4 cm, and, in some cases, the uterus, vagina, and perimeters might be described as well. The prognosis is unfavourable if the lymph nodes are involved [10].

PET-CT is credited as a highly positive and specific method for lymphoma stage determination [1]. From an imaging investigation point of view, PET-CT has become the method of choice for both diagnosis and patient evolution monitoring, sometimes in alternation with MRI [12,13,14]. We must mention that the cervical smear (Papanicolaou test) has low sensitivity for this pathology [15]. Therefore, advance cervical cancer screenings are important in early determination of this disease. Gynaecologists should know more about it and cooperate with anatomy and pathology doctors, as well as haematologists, in determining the diagnosis, with immunohistochemistry being essential for an early diagnosis, differential diagnosis, determination of the stage, and for initiating therapy.

## 4. Conclusions

Extranodal genital and cervical MNHLs are very rare, and often associated with late diagnosis. Combining histological and pathological investigation and immunohistochemistry leads to a precise cervical MNHL diagnosis and contributes to a differential diagnosis against other genital tumours. Imaging scans are important in diagnosing and monitoring this pathology (CT, MRI, PET-CT).

The main treatment is chemotherapy, with radiotherapy as an adjuvant. If found early and treated, the prognosis for this disease is very good, with an 80% 5-year survival rate.

## Figures and Tables

**Figure 1 medicina-58-00106-f001:**
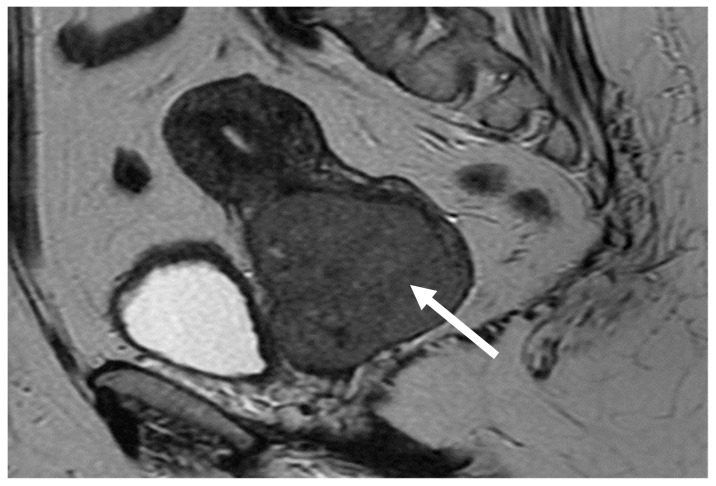
Lymphomatous cervical tumour invading the vaginal cavity: MRI (Magnetic Resonance Imaging) scan.

**Figure 2 medicina-58-00106-f002:**
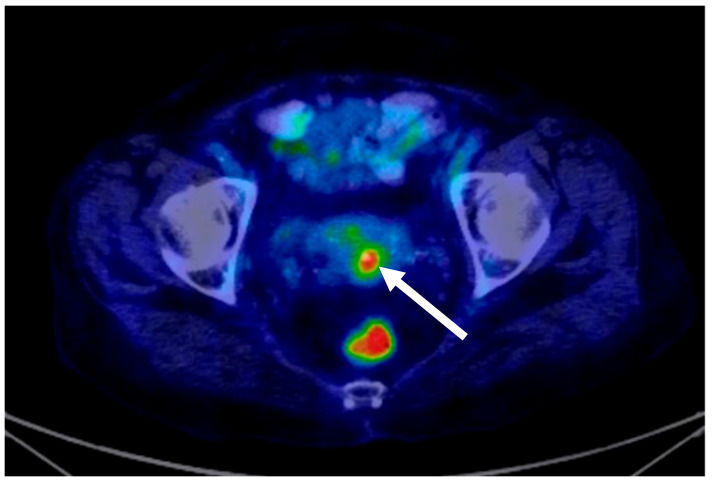
Residual lymphomatous cervical tumour post-chemotherapy, six rounds: PET-CT (Positron Emission Tomography—Computed Tomography) scan.

**Figure 3 medicina-58-00106-f003:**
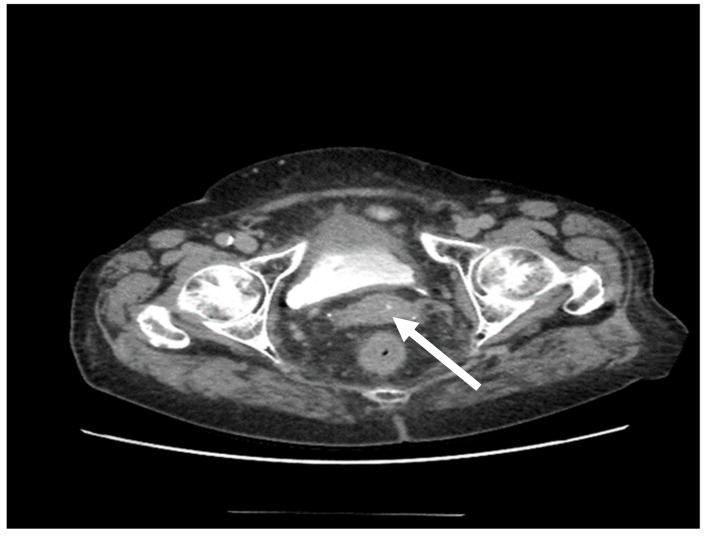
The 29-month CT scan, normal-looking cervix, post-chemotherapy and post-radiotherapy for a uterine cervix primitive non-Hodgkin lymphoma.

**Table 1 medicina-58-00106-t001:** Primitive uterine cervix lymphoma.

Authors	Article Year	No. of Cases/Patient Age (Years)	Immunohistochemistry	Histology and Pathology	Treatment	Disease-Free Time Interval
1. Contreras-Chavez P et al. [5]	2018	1/79	positive: CD20, BCL2, BCL6, CD10, MUM-1 (GC-B type), and C-MYC, Ki-67 in 70% negative: Pankeratin, Melan-A, S100, and CD3.	(DLBCL) (GC-B type),	6 × R-CHOP	12 months/CT
2. Mathilde Del et al. [8]	2020	1/36	positive: CD45, CD20 (B-cell marker), Bcl6, Bcl2, and MYC, and negative for CD3 (T-cell marker), CD10, MUM1 (GC-B type), CD5, and cyclin D1, with a Ki67 index (60%)	(DLBCL) (GC-B type)	6 × R-CHOP. Full response—tumour disappeared	15 months/clinical assessment + PET-CT
3. Jayant Sastri Goda et al. [10]	2020	4/average age 50 (39–62)	positive: CD20, CD10, Ki-67 index 80% negative: Mum-1 (multiple myeloma oncogene 1), and Bcl-6(B-cell lymphoma-6)	(DLBCL) GC-B type (2) Non-GC-B type (2 Molecular subtype (by Hans Algorithm))	6 × R-CHOP + RT (45Gy)	20 months (8–43)/clinical assessment + PET-CT/MRI
4. Fontana, C S. et al. [11]	2019	1/38	−	(DLBCL)	6 × R-CHOP + hysterectomy	12 months/PET-CT/CT
5. Seidler SJ et al. [12]	2018	1/50	positive: CD20 (B cell), CD3 (T cell), CD45, CD10, BCL-6, CD5, MUM-1(GC-B type), Cyclin D1, Ki67 negative: Bcl-2, MYC	(DLBCL) (GC-B type),	6 × R-CHOP	12 months/PET-CT/CT
6. Cubo AM et al. [13]	2017	1/51	positive: CD20+, CD5+, BCL2+, BCL6+, CD45+, CD23+, CD43+ negative: MUM1 (GC-B type), CD10, CD30, CyclinD1, and EBER. Ki-67 and p53 positive ≥ 60%	(DLBCL) (GC-B type),	6 × R-CHOP	24 months/PET-CT/MRI
7. Benedetta Desana et al. [14]	2020	1/54	-	(DLBCL)	6 × R-CHOP + radical hysterectomy	12 months/PET-CT/MRI
8. Guang Yang et al. [15]	2017	1/69	positive: CD45, CD20şi PAX-5, MUM1 (GC-B type), BCL2+, BCL6+ negative: CD10 Ki67 99%	(DLBCL) (GC-B type),	6 × R-CHOP	12 months/PET-CT/CT
9. Ana Regalo et al. [16]	2016	1/40	positive: for CD20, CD10, bcl2, and bcl6 negative: CD5, CD3, CD23, and cyclin D	(DLBCL)	8 × R-CHOP Recidivation after 45 months + 4 × R-CHOP + RT	45 months +3 months post-RT for recidivation/PET-CT/CT
10. Díaz De-La-Noval B [17]	2016	1/46	positive: CD20, CD45, BCL6, and CD30, 50% Ki67 negative: CK-AE1/AE3, CD15, AML, CD68, BCL2, CD10, MUM1, CD5, and C-MYC	(DLBCL) (non-GC-B type)	6 × R-CHOP	12 months/PET-CT
11. Weiyan Zhou [18]	2016	1/31	−	(DLBCL)	6 × R-CHOP	12 months/PET-CT/MRI

DLBCL = diffuse large B-cell lymphoma, GC-B = germinal centre-B, R-CHOP = rituximab plus cyclophosphamide, doxorubicin, vincristine, and prednisone, RT = radiotherapy, MRI = magnetic resonance imaging, CT = computer tomography, PET-CT = positron emission tomography–computer tomography.

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
