# Peer review of "Primary Non-Hodgkin Uterine Lymphoma of the Cervix: A Literature Review"

_medicina, 2022, doi:10.3390/medicina58010106_

Round 1

Reviewer 1 Report

Dear authors,

Thank you for the opportunity to allow me to review your work.

DLBCL of the cervix is rare, and this case resulted in an excellent outcome for this elderly patient.  Unfortunately, I do not believe there is sufficient novelty in this particular case to merit publication.  Although it is a unique site, DLBCL of the cervix has been reviewed in multiple prior publications, and I do not see anything novel in this particular case.

A few specific recommendations in case publication is sought elsewhere:

1) The literature review is incomplete.  Table 1 should be redone with each row representing a case series that has been reviewed -- however, make sure they are not double-counted since some series may include other older series.  Would include the breakdown by stage for each series, survival outcome, and maybe GC vs. non-GC if there.

2) The decision about adding rituximab or not to a HCV-positive pt with DLBCL is complicated.  https://ashpublications.org/blood/article/116/24/5119/27996/Hepatic-toxicity-and-prognosis-in-hepatitis-C Would describe the HCV more in detail -- what was the viral load?  What were the transaminases at baseline (normal/elevated)?  Any evidence of baseline cirrhosis?  

I. Introduction.

Remove the first sentence.  Of note, MNHL should not be used as an abbreviation as it is non-standard -- just use NHL (although I recommend the entire sentence be redacted).

Remove the third sentence -- would not describe GC vs. non-GC in the introductory paragraph.

Line 39 - change "primitive" to "primary"

Remove the first line of paragraph 3 -- delayed diagnosis is not necessarily the only reason that extranodal NHL fares worse and this statement is not entirely supported by the literature (in reality it is site-specific).  Better to omit it completely.

Line 44 - change "are represented by" to "may include"

LIne 46-47 - remove this first sentence -- should not be that difficult to diagnose once a biopsy is done.

Line 63 - remove the (ACL).  The abbreviation does not help and is not commonly used.

Line 63-64 - remove the line of the Hans algorithm.

Line 66 - change "...and no secondary..." to "and no evidence of marrow involvement was found."

Line 75 - Reword to "The diagnosis of stage IE DLBCL of the cervix was established"

Line 77-78 - Do you mean doxorubicin 50 mg/m2 instead of mg/mp?  Cyclophosphamide 600 mg/m2 instead of mg/day?  Cyclophosphamide is only given once every 21 days.  Please make sure units are correct, and the only multi-day drug is the steroid so it should be dexamethasone 16 mg/day x 5 days and mention it is given every 21 days.  This is important because this is not a standard CHOP prescription which prednisone / prednisolone is typically used.

Line 83 - Change "Control" to "post treatment"

Line 84 - change "with" to "consistent with"

Lines 84-86 - reword this sentence and remove the acronym since it is not used again.  I would say "Multidisciplinary consultation resulted in the recommendation for consolidation radiotherapy; 45 Gy was administered over 25 fractions.

Line 87 - "Two contrast-enhanced" instead of "contrast substance"

Line 88-89 - do you mean no regional adenopathy?

Line 90 - change "control" to surveillance" since this was done well after therapy completed

Lines 107-108 - I would remove the hepatitis C being correlated with DLBCL -- correlation is mild and I'm not sure it is particularly interesting unless it is for some reason correlated with cervix DLBCL

Lines 110-112 - 20% survival for elderly pts with advanced stage dz is a statistic I probably wouldn't include.  I would instead cite that this was a stage IE case.  What is the average stage of pts diagnosed with cervix DLBCL based on your literature review?  Are most pts diagnosed with localized or advanced disease?

Lines 113-114 - you mention 14 cases were those described in the literature; however, there are larger series that have been reviewed.  I would not dissect 14 individual cases.  For instance, https://ar.iiarjournals.org/content/34/8/4377 cites 144 cases.  It is more helpful to cite the case series.

Author Response

Response to Reviewer 1 Comments

Open Review

(x) I would not like to sign my review report

( ) I would like to sign my review report

English language and style

( ) Extensive editing of English language and style required

(x) Moderate English changes required

( ) English language and style are fine/minor spell check required

( ) I don't feel qualified to judge about the English language and style

Is the work a significant contribution to the field?   

Is the work well organized and comprehensively described?

Is the work scientifically sound and not misleading?

Are there appropriate and adequate references to related and previous work?         

Is the English used correct and readable?     

Comments and Suggestions for Authors

Dear authors,

Thank you for the opportunity to allow me to review your work.

DLBCL of the cervix is rare, and this case resulted in an excellent outcome for this elderly patient.  Unfortunately, I do not believe there is sufficient novelty in this particular case to merit publication.  Although it is a unique site, DLBCL of the cervix has been reviewed in multiple prior publications, and I do not see anything novel in this particular case.

A few specific recommendations in case publication is sought elsewhere:

1) The literature review is incomplete.  Table 1 should be redone with each row representing a case series that has been reviewed -- however, make sure they are not double-counted since some series may include other older series.  Would include the breakdown by stage for each series, survival outcome, and maybe GC vs. non-GC if there.

2) The decision about adding rituximab or not to a HCV-positive pt with DLBCL is complicated.  https://ashpublications.org/blood/article/116/24/5119/27996/Hepatic-toxicity-and-prognosis-in-hepatitis-C Would describe the HCV more in detail -- what was the viral load?  What were the transaminases at baseline (normal/elevated)?  Any evidence of baseline cirrhosis? 

  1. Introduction.

Remove the first sentence.  Of note, MNHL should not be used as an abbreviation as it is non-standard -- just use NHL (although I recommend the entire sentence be redacted).

Remove the third sentence -- would not describe GC vs. non-GC in the introductory paragraph.

Line 39 - change "primitive" to "primary"

Remove the first line of paragraph 3 -- delayed diagnosis is not necessarily the only reason that extranodal NHL fares worse and this statement is not entirely supported by the literature (in reality it is site-specific).  Better to omit it completely.

Line 44 - change "are represented by" to "may include"

LIne 46-47 - remove this first sentence -- should not be that difficult to diagnose once a biopsy is done.

Line 63 - remove the (ACL).  The abbreviation does not help and is not commonly used.

Line 63-64 - remove the line of the Hans algorithm.

Line 66 - change "...and no secondary..." to "and no evidence of marrow involvement was found."

Line 75 - Reword to "The diagnosis of stage IE DLBCL of the cervix was established"

Line 77-78 - Do you mean doxorubicin 50 mg/m2 instead of mg/mp?  Cyclophosphamide 600 mg/m2 instead of mg/day?  Cyclophosphamide is only given once every 21 days.  Please make sure units are correct, and the only multi-day drug is the steroid so it should be dexamethasone 16 mg/day x 5 days and mention it is given every 21 days.  This is important because this is not a standard CHOP prescription which prednisone / prednisolone is typically used.

Line 83 - Change "Control" to "post treatment"

Line 84 - change "with" to "consistent with"

Lines 84-86 - reword this sentence and remove the acronym since it is not used again.  I would say "Multidisciplinary consultation resulted in the recommendation for consolidation radiotherapy; 45 Gy was administered over 25 fractions.

Line 87 - "Two contrast-enhanced" instead of "contrast substance"

Line 88-89 - do you mean no regional adenopathy?

Line 90 - change "control" to surveillance" since this was done well after therapy completed

Lines 107-108 - I would remove the hepatitis C being correlated with DLBCL -- correlation is mild and I'm not sure it is particularly interesting unless it is for some reason correlated with cervix DLBCL

Lines 110-112 - 20% survival for elderly pts with advanced stage dz is a statistic I probably wouldn't include.  I would instead cite that this was a stage IE case.  What is the average stage of pts diagnosed with cervix DLBCL based on your literature review?  Are most pts diagnosed with localized or advanced disease?

Lines 113-114 - you mention 14 cases were those described in the literature; however, there are larger series that have been reviewed.  I would not dissect 14 individual cases.  For instance, https://ar.iiarjournals.org/content/34/8/4377 cites 144 cases.  It is more helpful to cite the case series.

Submission Date

14 December 2021

Date of this review

22 Dec 2021 15:17:31

Dear reviewer

Thank you for the advice and guidance given in making the article. Your advice has been very helpful in making this article.

DLBCL of the cervix is rare, and this case resulted in an excellent outcome for this elderly patient.  Unfortunately, I do not believe there is sufficient novelty in this particular case to merit publication.  Although it is a unique site, DLBCL of the cervix has been reviewed in multiple prior publications, and I do not see anything novel in this particular case.

DLBCL of the cervix is rare, in the "Alexandru Trestioreanu" Oncological Institute in Bucharest in 30 years of activity no case has been registered, although 25,000 cancer patients are treated annually. We did not find in our country (Romania) any article regarding this pathology. The article is not a case presentation, but is intended to be a review of the literature of recent years, where there is still no consensus in terms of treatment (chemotherapy alone, chemoradiotherapy?). Our case is characterized by the association with DCLB and hepatitis C that did not allow Rituximab, with a partial response to chemotherapy, and the need to combine irradiation. The result of the treatment was spectacular, the tumor disappearing on clinical examinations, PET-CT and histopathology. The patient is also living in the present moment, being without signs of locoregional and distant recurrence.

1.The literature review is incomplete.  Table 1 should be redone with each row representing a case series that has been reviewed -- however, make sure they are not double-counted since some series may include other older series.  Would include the breakdown by stage for each series, survival outcome, and maybe GC vs. non-GC if there.

  • The data presented in the table are the only ones I found in PubMed in the last 5 years (2016-2020) regarding cervical lymphoma (14 cases). The authors have at most 1-4 cases, present the way of diagnosis and treatment and compare the response to treatment with the results obtained by other authors. Unfortunately, there is no systematization of immunohistochemical tumor markers that need to be worked on to establish the diagnosis of aggression and prognosis of response to treatment.
  • I revised the table and where I found it I introduced the GC and nonGC classification according to the Hans algorithm
  1. I. Introduction.

Remove the first sentence.  Of note, MNHL should not be used as an abbreviation as it is non-standard -- just use NHL (although I recommend the entire sentence be redacted).

  • I deleted the first sentence from the Introduction
  1. Remove the third sentence -- would not describe GC vs. non-GC in the introductory paragraph
  • I remove the third sentence
  1. Line 39 - change "primitive" to "primary"

- I changed "primitive" to "primary"

  1. Remove the first line of paragraph 3 -- delayed diagnosis is not necessarily the only reason that extranodal NHL fares worse and this statement is not entirely supported by the literature (in reality it is site-specific). Better to omit it completely.

- I remove the first line of paragraph 3

  1. Line 44 - change "are represented by" to "may include"

- I change with may include

  1. LIne 46-47 - remove this first sentence -- should not be that difficult to diagnose once a biopsy is done.

- I removed the first sentence

  1. Line 63 - remove the (ACL). The abbreviation does not help and is not commonly used

- I removed the ACL

  1. Line 63-64 - remove the line of the Hans algorithm.

- I removed Hans algorithm

  1. Line 66 - change "...and no secondary..." to "and no evidence of marrow involvement was found."

- I changed in „and no evidence of marrow involvement was found."

  1. Line 75 - Reword to "The diagnosis of stage IE DLBCL of the cervix was established"

- I reformulated according to the recommendation

  1. Line 77-78 - Do you mean doxorubicin 50 mg/m2 instead of mg/mp? Cyclophosphamide 600 mg/m2 instead of mg/day? Cyclophosphamide is only given once every 21 days.  Please make sure units are correct, and the only multi-day drug is the steroid so it should be dexamethasone 16 mg/day x 5 days and mention it is given every 21 days.  This is important because this is not a standard CHOP prescription which prednisone / prednisolone is typically used.

 - Doxorubicin 50mg/m² at day 1, Vincristine 2mg/ m² at day 1, Cyclophosphamide 600mg/m² at day 1, Dexame-thasone 16mg/m² from day 1 to day 5 it is given every 21 days

  1. Line 83 - Change "Control" to "post treatment"

- I changed to post treatment

  1. Line 84 - change "with" to "consistent with"

- I changed to „consistent with”

  1. Lines 84-86 - reword this sentence and remove the acronym since it is not used again. I would say "Multidisciplinary consultation resulted in the recommendation for consolidation radiotherapy; 45 Gy was administered over 25 fractions.

- I made the recommended change

  1. Line 87 - "Two contrast-enhanced" instead of "contrast substance"

- I made the recommended change

  1. Line 88-89 - do you mean no regional adenopathy?

- Yes, no regional adenopathy. I modified and specified in the text

  1. Line 90 - change "control" to surveillance" since this was done well after therapy completed

- I changed "control" to surveillance"

  1. Lines 107-108 - I would remove the hepatitis C being correlated with DLBCL -- correlation is mild and I'm not sure it is particularly interesting unless it is for some reason correlated with cervix DLBCL

- I deleted the paragraph with the correlation between Hepathy C and DCLB

  1. Lines 110-112 - 20% survival for elderly pts with advanced stage dz is a statistic I probably wouldn't include. I would instead cite that this was a stage IE case. What is the average stage of pts diagnosed with cervix DLBCL based on your literature review?  Are most pts diagnosed with localized or advanced disease?

- This is the corrected paragraphThe case we’re presenting is an IE Ann Arbor stage (6,7). Studies indicate that 5-year survival is better in the IE stage (89%), than in the IIE - IV stages (20%). (2)

  1. Lines 113-114 - you mention 14 cases were those described in the literature; however, there are larger series that have been reviewed. I would not dissect 14 individual cases. For instance, https://ar.iiarjournals.org/content/34/8/4377 cites 144 cases.  It is more helpful to cite the case series.
  • The article presents 14 cases of lymphoma of the uterine cervix published in the literature during 2016-2020. Indeed, there are large batch studies such as the one in ANTICANCER RESEARCH 36: 4931-4940 (2016), doi: 10.21873 / anticancer.11059 which includes 246 cases of everything that has been published about Lymphoma of the uterine cervix. The exemplified study https://ar.iiarjournals.org/content/34/8/4377 includes 143 cases from January 1990 to June 2013 for English articles and abstracts showing data on primary DLBCL of the uterus. Probably in the 2 articles some of these cases overlap. Through my work I wanted to avoid this
  • Please see the attachment

Reviewer 2 Report

  1. More details should be added in the text on the role of FDG PET: did the patient perform PET/CT at staging? Did the patient perform PET/CT at interim? What the 5-point scale of DEAUVILLE score was at restaging PET/CT?
  2. More details should be added in the text regarding the CHOP-21 schedule and the R-CHOP-21 schedule
  3. More details should be added in the text regarding the HCV infection (such as serum levels of HCV-RNA, HCV-genotypes). Did the patient perform antiviral therapy to eradicate the infection? What the liver damage was (active chronic hepatitis or cirrhosis)? Is really rituximab not indicated in patients with NHL and HCV infection?
  4. Please, remove “malignant” before non-Hodgkin lymphoma. It is better Hodgkin lymphoma instead of Hodgkin disease
  5. Use of English language should be improved in all the text and table
  6. In Discussion section, the authors should state that nowadays there are potent antiviral drugs against HCV-infections in patients with lymphomas.

Author Response

Response to Reviewer 1 Comments

Open Review

(x) I would not like to sign my review report

( ) I would like to sign my review report

English language and style

(x) Extensive editing of English language and style required

( ) Moderate English changes required

( ) English language and style are fine/minor spell check required

( ) I don't feel qualified to judge about the English language and style

Is the work a significant contribution to the field?   

Is the work well organized and comprehensively described?

Is the work scientifically sound and not misleading?

Are there appropriate and adequate references to related and previous work?         

Is the English used correct and readable?     

Comments and Suggestions for Authors

More details should be added in the text on the role of FDG PET: did the patient perform PET/CT at staging? Did the patient perform PET/CT at interim? What the 5-point scale of DEAUVILLE score was at restaging PET/CT?

More details should be added in the text regarding the CHOP-21 schedule and the R-CHOP-21 schedule

More details should be added in the text regarding the HCV infection (such as serum levels of HCV-RNA, HCV-genotypes). Did the patient perform antiviral therapy to eradicate the infection? What the liver damage was (active chronic hepatitis or cirrhosis)? Is really rituximab not indicated in patients with NHL and HCV infection?

Please, remove “malignant” before non-Hodgkin lymphoma. It is better Hodgkin lymphoma instead of Hodgkin disease

Use of English language should be improved in all the text and table

In Discussion section, the authors should state that nowadays there are potent antiviral drugs against HCV-infections in patients with lymphomas.

Submission Date

14 December 2021

Date of this review

Dear reviewer

Thank you for the advice and guidance given in making the article.

Comments and Suggestions for Authors

1.More details should be added in the text on the role of FDG PET: did the patient perform PET/CT at staging? Did the patient perform PET/CT at interim? What the 5-point scale of DEAUVILLE score was at restaging PET/CT?

- PET-CT was used for post-therapeutic monitoring according to NCCN Version 5.2021 B-Cell Lymphomas which can be used simultaneously or separately with CT.

- I entered in the text Lugano category 2: score 4 scale of Deauville five-point scale) (1) with reduction of intensity from baseline, but no new lesions and Lugano category 1: score 1,2,3 of scale of Deauville five-point scale likely to represent complete metabolic response after radiotherapy

  1. More details should be added in the text regarding the CHOP-21 schedule and the R-CHOP-21 schedule

- I introduced the changes to CHOP-21 in the text. and the R-CHOP-21 schedule

  1. More details should be added in the text regarding the HCV infection (such as serum levels of HCV-RNA, HCV-genotypes). Did the patient perform antiviral therapy to eradicate the infection? What the liver damage was (active chronic hepatitis or cirrhosis)? Is really rituximab not indicated in patients with NHL and HCV infection?

- Yes, the patient was diagnosed with chronic hepatitis C (diagnosis of HCV-RNA = 134,000 i.u./l). I have completed it in the text. The patient did not take antiviral therapy but went to see an infection doctor before starting chemotherapy. According to NCCN Clinical Guideline version 5.2021, Rituximab is not recommended as it may induce liver failure. Rituximab being removed from the therapy under very specific conditions (e.g. chronic hepatitis, etc.), with radiotherapy as an adjuvant treatment and surgery (full hysterec-tomy) being used in a small number of pre- or post-chemotherapy cases (I mentioned this in the Discussions) See also the references: https://doi.org/10.1016/j.gore.2020.100639, https://doi.org/10.1002/cnr2.1264

  1. Please, remove “malignant” before non-Hodgkin lymphoma. It is better Hodgkin lymphoma instead of Hodgkin disease

- I made the suggested changes

  1. Use of English language should be improved in all the text and table

 - Yes. I sent to the editorial office magazines for the correct translation into English (English editing ID: english-38736)

  1. In Discussion section, the authors should state that nowadays there are potent antiviral drugs against HCV-infections in patients with lymphomas.

- I wrote that according to the therapeutic guidelines (for example NCCN) antiviral therapy against HVC can be used before starting chemoimmunotherapy and can thus be integrated into the treatment plan.

Round 2

Reviewer 1 Report

accept -- would do revisions as per reviewer #2

Author Response

Thanks again for the tips and guidanceThanks again for the tips and guidance

Reviewer 2 Report

The authors have stated that PET-CT was used for post-therapeutic monitoring according to NCCN Version 5.2021 and that according to NCCN Clinical Guideline version 5.2021, Rituximab is not recommended as it may induce liver failure. However, the patient was diagnosed and treated in the 2018. The authors should explain how have used the guidelines of 2021 in a clinical case of 2018.

The authors now report "the text Lugano category 2: score 4 scale of Deauville five-point scale) (1) with reduction of intensity from baseline, but no new lesions and Lugano category 1: score 1,2,3 of scale of Deauville five-point scale likely to represent complete metabolic response after radiotherapy". I think that it is better to report in the text only the number of the score of the Deauville five-point scale for interpreting FDG PET scans.

Author Response

I would not like to sign my review report

( ) I would like to sign my review report

English language and style

( ) Extensive editing of English language and style required

( ) Moderate English changes required

(x) English language and style are fine/minor spell check required

( ) I don't feel qualified to judge about the English language and style

Is the work a significant contribution to the field?   

Is the work well organized and comprehensively described?

Is the work scientifically sound and not misleading?

Are there appropriate and adequate references to related and previous work?         

Is the English used correct and readable?     

Response to Reviewer:

Thanks again for the tips and guidance

Comments and Suggestions for Authors

1.The authors have stated that PET-CT was used for post-therapeutic monitoring according to NCCN Version 5.2021 and that according to NCCN Clinical Guideline version 5.2021, Rituximab is not recommended as it may induce liver failure. However, the patient was diagnosed and treated in the 2018. The authors should explain how have used the guidelines of 2021 in a clinical case of 2018.

- Thank you. The treatment and monitoring of the patient was performed according to the clinical guidelines of NCCN 2018. Indeed, when I wrote the article I checked the 2021 guide.

- I made changes to the bibliography. I passed the NCCN clinical guide from 2018. At least there are no changes in the way the patient is treated and monitored. (NCCN Guidelines for B-cell Lymphomas V2.2018) (https://oncolife.com.ua/doc/nccn/B-Cell_Lymphomas.pdf)

- Regarding chemoimmunotherapy, the 2018 guide states that it can be performed with careful monitoring of the patient and liver function (clinical, tests, etc.), and antiviral therapy can be performed after the end of cancer treatment. I would like to mention that in our case the chemotherapy was started urgently, after establishing the diagnosis of lymphoma, because the patient was bleeding from the cervix tumor and had anemia. Before starting the chemotherapy treatment I had the consultation of the infectionist doctor regarding HVC

The authors now report "the text Lugano category 2: score 4 scale of Deauville five-point scale) (1) with reduction of intensity from baseline, but no new lesions and Lugano category 1: score 1,2,3 of scale of Deauville five-point scale likely to represent complete metabolic response after radiotherapy". I think that it is better to report in the text only the number of the score of the Deauville five-point scale for interpreting FDG PET scans.

  • I made the proposed changes to the score of the Deauville
